# Non-Alcoholic Fatty Liver Disease in Poultry: Risk Factors, Mechanism of Development, and Emerging Strategies

**DOI:** 10.3390/ijms26178460

**Published:** 2025-08-30

**Authors:** Aneeqa Imtiaz, Muhammad Talha Bin Tahir, Minmeng Zhao, Daoqing Gong, Jing Ge, Tuoyu Geng

**Affiliations:** 1College of Animal Science and Technology, Yangzhou University, Yangzhou 225009, China; aneeqaimtiaz342@yahoo.com (A.I.); talhatahir2640@gmail.com (M.T.B.T.); zhaominmeng123@163.com (M.Z.); yzgong@163.com (D.G.); 2Key Laboratory for Animal Genetics, Breeding, Reproduction and Molecular Design of Jiangsu Province, College of Animal Science and Technology, Yangzhou University, Yangzhou 225009, China; 3Joint International Research Laboratory of Agriculture and Agri-Product Safety of the Ministry of Education of China, Yangzhou University, Yangzhou 225009, China

**Keywords:** non-alcoholic fatty liver disease, lipid accumulation, nutritional intervention, genetic selection, environmental stress, laying hen

## Abstract

Non-alcoholic fatty liver disease (NAFLD) has emerged as a significant metabolic disorder in modern poultry production, particularly affecting high-yielding laying hens. This condition compromises bird welfare, productivity, and economic sustainability within commercial farming systems. This narrative review provides a comprehensive overview of the underlying mechanisms through which hepatic lipid accumulation, metabolic dysfunctions, hormonal imbalances, genetic susceptibilities, and environmental stress contribute to the development of NAFLD. The multifactorial nature of NAFLD is explored through a critical assessment of the literature, highlighting the influence of diet composition, management practices, and physiological demands associated with intensive egg production. Emphasis is placed on recent advancements in nutritional modulation, selective breeding, and housing improvements aimed at prevention and mitigation of NAFLD. Furthermore, the review identifies key research gaps, including limited understanding of epigenetic influences and the long-term efficacy of intervention strategies. An integrative framework is advocated, synergizing genetics, nutrition, and environmental optimization to effectively address the complexity of NAFLD in poultry and supports the development of resilient production systems. The insights presented aims to inform both future research and practical applications for enhancing poultry health and performance.

## 1. Introduction

Non-alcoholic fatty liver disease (NAFLD) is a globally prevalent metabolic disorder that has gained considerable attention due to its increasing incidence and potential to progress into severe hepatic complications [1]. In humans, NAFLD encompasses a disease spectrum ranging from simple steatosis to non-alcoholic steatohepatitis (NASH), fibrosis, cirrhosis, and hepatocellular carcinoma, without a history of excessive alcohol consumption [2]. NAFLD is characterized by fat accumulation in hepatocytes or hepatic steatosis [3]. Hepatic steatosis affects 38% of human population globally. Its prevalence can reach 70% among type 2 diabetes mellitus (T2DM) groups and even affects 7% of individuals with no apparent health problems [4]. Recent epidemiological data suggest that NAFLD has a high incidence rate reported in the Middle East (32%) and South America (30%) [5]. Strongly associated with obesity, insulin resistance, type 2 diabetes, and sedentary lifestyles, NAFLD has become a leading cause of chronic liver disease and imposes a significant burden on public health systems [6].

A similar pathological condition is increasingly recognized in avian species, particularly in commercial poultry. NAFLD in poultry, especially among high-yielding laying hens, is characterized by the excessive accumulation of triglycerides within hepatocytes, leading to hepatic dysfunction, reduced productivity, and increased mortality [7]. Traditionally regarded as an incidental or minor condition, NAFLD in poultry is now acknowledged as a serious threat to animal welfare and the economic viability of intensive production systems [7].

Fatty liver hemorrhagic syndrome (FLHS) is a global problem in laying hens, observed worldwide [8]. Baseline surveys indicate that a few percent of hens are typically affected at any time, but incidence rates can vary widely. One study estimated, laying flocks had an average prevalence of about 5%, with occasional outbreaks affecting at least 15% to 20% of the birds [9,10].

In caged, high-production layer flocks, which are especially prone due to overfeeding and restricted movement, FLHS has become one of the leading causes of noninfectious disease in layer hens [11]. For example, in extremely affected birds, FLHS is responsible for over 70% of flock mortality in a region (e.g., in Queensland, Australia) [12]. The high mortality in affected flocks represents a direct economic loss due to a fall in level of future egg production. Researchers have observed that FLHS-prone hens tend to have abnormally high feed intake and body weight, followed by a sharp decline in egg output during the laying cycle [12].

The poultry sector produces 82 million Metric Tons (FAO) of eggs for total world production, with Asia being the major contributor. As eggs are a major source of various nutritional factors like vitamins, proteins, and essential fats like omega-3, these factors underline the importance of the commercial layer sector to elucidate the global malnutritional problems [13]. Layers are at a higher risk of developing NAFLD due to high energy diets fed during late laying periods, causing substantial economic and productivity losses [14].

Notably, fatty liver hemorrhagic syndrome (FLHS), a severe form of NAFLD is implicated in up to 74% of mortalities among laying hens kept in cage system during peak egg production periods [15]. The etiology of the disease is multifactorial, and various risk factors are associated with the onset of the disease including dietary factors, genetic susceptibility, gut physiology, physical health condition, and environmental stress such as high temperature and poor housing conditions (Table 1). Obesity, insulin resistance, and cardiovascular risks are generally characteristics of the metabolic syndrome, which are often connected to NAFLD [16].

Amino acids are one of the dietary factors actively involved in more fat deposition due to impaired fatty acid synthesis and utilization. Molecular studies reveal oxidative stress, mitochondrial dysfunction, and the release of inflammatory cytokines are contributing factors to the development of NAFLD and worsen liver cell damage [17]. The proper function of cellular membranes and the control of lipid metabolism depend severely on essential fatty acids, including omega-3 and omega-6 [18]. The integration of improper fatty acids causes breakdown of metabolic pathways, which subsequently leads to hepatic fat accumulation in the liver cells [19].

Endocrine regulation serves as an essential factor through the activities of estrogen hormones [20]. The estrogen surges in laying hens lead to hepatic lipids and yolk protein synthesis, which helps the birds produce eggs [21,22]. Insulin resistance poses an extra challenge to liver cells as it leads to reduced glucose uptake combined with elevated De novo lipogenesis that intensifies hepatic lipid buildup [23].

The development of NAFLD in chickens can also be intensified by environmental conditions such as heat stress and overcrowding [24]. Excessive heat forces multiple adverse effects on hepatic metabolism through its generation of oxidative damage combined with suppressed immunity and endocrine signal interruption [25]. The feeding patterns of heat-stressed birds become abnormal, and their feed consumption decreases which surprisingly triggers the release of higher amounts of stored fat that result in additional strain on the liver [26]. Pesticides, industrial solvents, and particulate matter are important environmental contaminants associated with the development of NAFLD [27]. Lead Posterior Inclusion Probability (PIP) of 1.000, as a significant contributor to NAFLD risk, shows a positive correlation with liver injury markers [28].

The incidence of NAFLD in poultry has been documented in various regions worldwide, highlighting its growing significance as a production and welfare concern [29]. Adhering to welfare guidelines and exploring non-animal methods are crucial to balancing scientific progress with ethical responsibility [30]. The practice of welfare-oriented farming reduces stress along with supporting natural behaviors, helping maintain both physiological equilibrium and strong immune functions, thus indirectly benefiting liver health [31].

**Table 1 ijms-26-08460-t001:** Risk factors associated with the onset of NAFLD in poultry.

Risk Factor	Impact	Reference
Dietary Factors	Energy-dense and high-carbohydrate diets, while intended to support high egg yield, contribute to excessive hepatic fat accumulation when not nutritionally balanced. Moreover, factors like overfeeding, vitamin deficiency, and mycotoxins are also associated with liver fattening conditions in poultry.	[32]
prolonged metabolic disease	Birds with prolonged metabolic disease patterns show severe hepatic pathologies such as fibrosis, cirrhosis, or hepatic failure, which can ultimately lead to mortality in layers.	[33]
Environmental Stress	Adverse conditions such as heat stress, poor ventilation, and overcrowding exacerbate metabolic disturbances and promote lipid deposition in the liver.	[34]
Gut Microbiota	In chickens, dysbiosis of the gut microbiota increases intestinal permeability, allowing the translocation of endotoxins such as lipopolysaccharide (LPS) into the blood stream. This promotes hepatic inflammation, disrupts lipid metabolism, and contributes to the progression of NAFLD.	[31,35]
Metabolic and Inflammatory Factors	Metabolic factors such as amino acid imbalance and insulin resistance in poultry are linked to NAFLD, whereas oxidative stress and ROS have a strong association with inflammatory pathways leading to NAFLD.	[36,37,38]
Genetic Predispositions	Different poultry breeds and genetic selection of high-producing layers and fast-growing broilers, genetically selected for traits like rapid growth or high egg production, leads to increased metabolic demand, hormonal imbalances, and a greater tendency for hepatic lipogenesis, making them more prone to fat accumulation in the liver.	[39]

To create this narrative review, a comprehensive search of the literature was performed using PubMed, Google scholar, and Web of Science. We found relevant articles by combining keywords like “non-alcoholic fatty liver disease” AND “poultry,” “hepatic steatosis” AND “laying hens” AND “lipid metabolism,” “fatty liver hemorrhagic syndrome” AND “mitochondrial function” AND “chickens,” “epigenetics” AND “NAFLD” AND “avian species,” and “nutritional intervention” AND “genetic selection” AND “hepatic lipid accumulation” AND “poultry health.” Furthermore, authoritative websites of esteemed organizations (e.g., FAO) were referenced for statistical data, production reports, and management recommendations. Two researchers independently performed the searches, eliminating duplicate records prior to screening. Eligible sources encompassed peer-reviewed experimental studies, epidemiological surveys, and authoritative reviews that offered empirical data or mechanistic insights into NAFLD risk factors, pathogenesis, and prevention or mitigation strategies in poultry.

## 2. Lipid Metabolism in Poultry

The poultry industry prioritizes egg and meat production, with lipid metabolism playing a vital role, particularly in laying hens [40]. In layers, excessive fat deposition adversely affects metabolic health and reproductive performance. Lipid accumulation is influenced by dietary energy, protein, amino acids, and mineral levels. Fats and oils, being dense energy sources, improve nutrient absorption, reduce heat increment, and enhance feed palatability [41].

The liver, a central organ in avian lipid metabolism, is primarily responsible for De novo fatty acid synthesis, lipid packaging, bile acid synthesis and transport, storage, oxidation, and ketone production when glucose is limited [42]. In laying hens, the extent of fat deposition depends on both dietary lipid intake and hepatic lipogenesis. Modulating dietary fat levels can influence egg yolk cholesterol content, impacting poultry physiology [43].

The significant similarity in lipid metabolism between humans and poultry suggests that both human and poultry NAFLD share similar mechanism of lipogenesis and fat deposition in the liver [44]; however, the liver of poultry shows some unique differences compared to mammals (Table 2). Apart from the fact that both poultry and mammals convert glucose to fatty acids with the help of acetyl-CoA carboxylase (ACC), fatty acid synthase (FAS), Lipase, and hormones like insulin and glucagon, they have different sites of lipogenesis. In poultry, liver serves as the main site of primary lipid metabolism, whereas in mammals, particularly in ruminants, adipose tissues serve as the primary site of lipogenesis. The difference also lies in lipoprotein transport and fatty acid composition [45,46]. In the primary catabolic pathway, fatty acid oxidation plays a crucial role, and several enzymes involved in this pathway are less regulated in hepatic steatosis [15].

## 3. Bile Acid Metabolism in Poultry

Avian species also demonstrate a distinct bile acid metabolism primarily in composition and function of bile acid, which significantly influences hepatic lipid homeostasis [53]. Bile acids synthesized in the liver are critical for emulsifying dietary fats and regulating lipid digestion [54].

Emulsified fats are broken down into fatty acids and monoglycerides. In poultry, pancreatic Lipase and colipase in intestinal epithelium breakdowns the fat molecules to form portomicrons after combining with cholesterol and fat-soluble vitamins. The absorbed portomicrons are released into hepatic portal vein from where they are taken up by hepatocytes for energy metabolism, and the excess are repackaged into VLDLs and fat droplets [55,56].

## 4. Mechanisms of Fatty Liver Development in Poultry

The liver, as the metabolic center for lipid metabolism in birds, is prone to steatosis due to high-calorie diets, hormone dysregulation, or environmental stresses like heat and overcrowding [57].

### 4.1. Role of Nutritional Imbalance in Development of NAFLD

The development of NAFLD in poultry is significantly influenced by nutrition. Lipid metabolism primarily occurs in the liver, and hepatic fat deposition can be directly impacted by dietary energy, protein, and essential nutrient imbalances. Birds are at risk for hepatic steatosis and FLHS when their diets lack essential lipotropic components like methionine, choline, and S-adenosylmethionine, which hinder lipid export and methylation reactions.

Methionine is an essential amino acid and a key methyl donor in one-carbon metabolism. In poultry, methionine deficiency is a well-known nutritional cause of fatty liver (analogous to NAFLD). Birds fed methionine-deficient diets show increased hepatic fat accumulation and even signs of liver injury [58,59]. Methionine shortfall impairs the synthesis of phosphatidylcholine (PC) for VLDL assembly and lowers VLDL secretion, causing triglycerides to accumulate in hepatocytes [60]. In addition, methionine is required for the synthesis of carnitine and cysteine; a deficiency can downregulate carnitine palmitoyltransferase-1 (CPT1), the rate-limiting enzyme in mitochondrial β-oxidation. With no methionine, the liver cannot export or burn fat effectively, promoting NAFLD development in poultry.

S-adenosylmethionine (SAMe), formed from methionine and ATP in the liver, is the principal methyl donor for critical reactions (including PC synthesis via the PEMT pathway) and a precursor for glutathione [61]. A recent multi-omics study in laying hens found that birds with severe fatty liver had markedly lower hepatic expression of methionine cycle enzymes (MAT1A, PEMT, AHCY, MTR), and an imbalance of related metabolites [60]. Conversely, maintaining adequate SAMe (via sufficient dietary methionine or methyl donors) supports proper methylation reactions and protects the liver from fat accumulation [61]. In summary, alterations in methionine/SAMe metabolism—whether through insufficient methionine intake, impaired conversion to SAMe, or lack of choline/methyl donor—play a pivotal role in poultry NAFLD by hampering the liver’s ability to package and utilize fat.

### 4.2. Role of Hepatic Lipid Accumulation and Bile Acid Dysregulation in NAFLD

NAFLD in poultry is primarily characterized by the excessive accumulation of lipids, especially triglycerides, within hepatocytes [62]. This disorder arises from metabolic imbalances involving disrupted lipogenesis, fatty acid oxidation, and lipoprotein secretion [63].

Hepatic triglyceride accumulation impairs normal liver function and can lead to progressive metabolic liver damage [64]. Laying hens require their livers to create crucial yolk precursors made of vitellogenin together with very low-density lipoproteins (VLDLs), which are necessary for oocyte development [65,66].

The lipotropic molecules methionine and choline help birds generate phospholipids to form VLDLs. The accumulation of lipids is intensified when birds suffer from nutrient deficiency, which leads to increased susceptibility to fatty liver development [67].

The high-calorie carbohydrate and fat-rich diet consumption by hens lead to increased liver lipogenesis while showing a reduced ability to oxidize lipids [68]. An accumulation of high amounts of lipids in the liver leads to ballooning hepatocytes along with inflammatory reactions and ultimately results in hepatic fibrosis when the condition persists [69]. Hepatic dysfunction occurs during this pathological process because it prevents necessary metabolic functions and impairs immune responses and endocrine processes for maintaining egg production [70].

Bile acid dysregulation plays a synergistic role in exacerbating hepatic lipid overload [71]. Impaired bile acid synthesis or transport can disrupt fat emulsification and absorption in the intestines [72], thereby altering enterohepatic circulation and increasing lipid retention in the liver. However, dysregulation of bile acid synthesis or secretion can impair lipid clearance from the liver and promote hepatic fat accumulation [73]. Cholesterol homeostasis is regulated by bile acid in the liver. Imbalance in bile acid regulation leads to fats deposition in hepatocytes, consequently increasing SREBP-1c (sterol regulatory element) expression and DNL, which, when prolonged, causes NAFLD in chickens [54]. The mRNA level of Farnesoid X Receptor (FXR) is suppressed in the liver and terminal ileum of hens with fatty liver, suggesting that FXR suppression disrupts lipid homeostasis, contributing to lipid accumulation and heightened NAFLD risk [74]. Studies have shown that disturbed bile acid signaling pathways, especially involving the FXR, exacerbate the progression of NAFLD in poultry [75].

Furthermore, dysbiosis of gut microbiota, a condition increasingly documented in intensively farmed poultry, can also influence bile acid profiles which further contribute to the onset of NAFLD [54].

### 4.3. Molecular Basis of NAFLD Development in Poultry

The molecular basis of fatty liver development in poultry revolves around identifying key genes and signaling pathways that govern lipid metabolism, oxidative stress, and energy balance [76,77]. According to a study conducted by Xiao et al. (2024) [78], 953 differentially expressed genes (DEGs) were linked to fatty liver disease in chickens, including 26 miRNAs, 56 lncRNAs, and key components of competing endogenous RNA (ceRNA) network. A significant lncRNA-miRNA-mRNA interaction involving ENSGALT00000079786 was highlighted in disease regulation [78,79]. Among the most frequently studied miRNAs, miR-216 and miR-217 are small noncoding RNAs often involved in liver fibrosis and liver-related metabolic disorders. In particular, they are often seen as differentially expressed in non-alcoholic fatty liver disease progression [78,79].

#### 4.3.1. Molecular Markers in NAFLD in Poultry

The development of NAFLD in poultry is intricately linked to various molecular markers that regulate lipid metabolism, inflammatory responses, and liver physiology. The expression of pro-inflammatory cytokines, such as TNF-α, IL-6, and IL-1β, was markedly elevated in the liver and abdominal adipose tissue of hens subjected to the HELP diet [67]. Elevated inflammatory marker levels are frequently associated with the onset of NAFLD [80]. A recent investigation was performed to identify the potential biomarkers for NAFLD in laying hens, which indicates that the concentrations of plasma acetoacetyl-CoA synthetase (AACS), dipeptidyl-peptidase 4 (DPP4), glutamine synthetase (GLUL), and glutathione S-transferase (GST) can serve as biomarkers for NAFLD. The chosen indicators exhibited substantial positive correlations with hepatic lipid accumulation (*p* < 0.001) [81].

#### 4.3.2. Signaling Pathways

Two pivotal signaling pathways in this context are AMP-activated protein kinase (AMPK) and peroxisome proliferator-activated receptor alpha (PPARα).

AMPK pathway: AMPK serves as an energy sensor, promoting lipid catabolism and inhibiting lipid synthesis. Activation of AMPK improves insulin sensitivity and reduces hepatic lipid accumulation [82]. Conversely, diminished AMPK activity, often resulting from metabolic disturbances, contributes to the progression of NAFLD. Emerging research suggests that dietary interventions activating the AMPK pathway can mitigate fatty liver development in laying hens [83].

PPARα signaling: PPARα functions as a nuclear receptor that governs fatty acid oxidation and lipid metabolism in poultry [84,85]. Activation of PPARα enhances hepatic β-oxidation of fatty acids, thereby reducing lipid accumulation [86]. Disruptions in PPARα signaling can impair lipid clearance, leading to hepatic steatosis [87]. Recent studies have highlighted the role of PPARα along with AMPK in modulating lipid metabolism and its potential as a therapeutic target in managing NAFLD in poultry [88].

### 4.4. The Role of ER-Mitochondrial Interplay in NAFLD

A critical component of NAFLD pathogenesis involves organelle-specific dysfunction, particularly the interaction between the endoplasmic reticulum (ER) and mitochondria [89]. This crosstalk plays a pivotal role in cellular homeostasis, and its disruption contributes significantly to hepatic pathology.

Mitochondrial dysfunction: Mitochondria, as the powerhouse of the cell, are involved in lipid metabolism. Reactive oxygen species (ROS) are generated within the mitochondria during lipid metabolism. The excessive accumulation of ROS enhances the permeability of the mitochondrial outer membrane (MOMP), diminishes mitochondrial membrane potential (MMP), and ultimately compromises mitochondrial integrity [90]. Impaired mitochondria engage in mitochondrial autophagy (mitophagy), which is essential in non-alcoholic steatohepatitis (NASH) via modulating lipid droplet accumulation [91]. The mitochondrial membrane integrity becomes compromised under stress conditions, resulting in impaired β-oxidation of fatty acids [92].

Endoplasmic reticulum (ER) stress and unfolded protein response (UPR): Lipid accumulation in hepatocytes induces ER stress, triggering the unfolded protein response (UPR) [93]. Persistent activation of UPR leads to upregulation of pro-apoptotic factors such as C/EBP Homologous Protein (CHOP) and markers like H2AX, which contribute to hepatocellular damage [94].

### 4.5. Genetic Predispositions of NAFLD in Poultry

Excessive and continuous genetic selection has led the poultry population to a high risk of fat accumulation and other metabolic disorders [95]. The threshold of commercial strains is lower than that of domestic and backyard breeds. For example, Leghorn chickens (layers) exhibit enhanced mobilization of n-3 polyunsaturated fatty acids (PUFAs) and triacylglycerols, whereas Ross chickens (broiler) prioritize lipid storage and structural functions [47]. This variation is driven by differential gene expression and enzymatic activity, particularly involving the FADS2 gene and Δ6-desaturase enzyme, which are more active in Leghorns, facilitating the synthesis of long-chain n-3 PUFAs [96].

Previous research has provided strong evidence of breed-specific variations, indicating that certain lines are naturally more prone to hepatic fat formation [79]. White Leghorn (WL) laying hens or Bei Jing You (BJY) local breeds were less likely to develop fatty liver than Jing Xing Huang (JXH) chickens (a dwarf broiler line) under the same feeding patterns. He found that the F_0_ males’ fatty liver phenotype was inherited to the F_1_ offspring, exhibiting enduring alterations in the expression of genes related to lipid metabolism (ACACA, FASN, SCD, ACSL5, FADS2, FABP1, and APOA4), implying that both genetic and epigenetic inheritance jointly affect the risk of NAFLD in offspring [79]. Additionally, quantitative genetic analyses in chickens revealed that hepatic steatosis in chickens showed relatively high heritability (h^2^ = 0.25) and duodenal microbiome heritability (m^2^ = 0.26), which links to reduced genetic capacity of hepatocytes to export triglycerides and fatty acids peroxidation [60].

Genetic predisposition factors for the development of NAFLD are often linked with nutritional or environmental factors [21]; however, the long-term effects of selecting specific breeds (especially in layers) prone to NAFLD are still limited. There is strong evidence suggesting that selection leads to liver fat accumulation in broiler experimental strains [32].

### 4.6. Role of Hormonal Dysregulation in Progression of NAFLD

Hormones along with nutritional factors are a leading cause of fatty liver development [97]. Estrogen plays a critical role in hepatic lipid metabolism and reproduction in poultry. It works by regulating vitellogenin synthesis and triglyceride transport during oogenesis. Imbalances, such as elevated estrogen levels in FLHS, disrupt lipid metabolism ultimately causing hepatic fat accumulation [98].

Estradiol (E2), the primary form of estrogen, plays a crucial role in the development of conditions like hypercholesterolemia and hypertriglyceridemia, and this mechanism leads to a pathway responsible for developing fatty liver condition through estrogen hormone due to deposition of lipids. Further correlation value (r = 0.96; *p* = 0.001) between white blood count and plasma fibrinogens concentration indicates estrogen responsible for liver damage [99].

The fundamental cause of the development of NAFLD in poultry is linked to insulin resistance, a metabolic condition in which the body’s ability to respond to insulin is reduced [100]. Insulin resistance results in overproduction in the liver of glucose and free fatty acids that are disposed towards fat accumulation in poultry. This condition is most frequently found in high-yielding commercial laying hens where the high metabolic demand of egg production adds to insulin resistance and worsens the chances of developing NAFLD [101,102]. Chronic low-grade inflammation, driven by TNF-α, promotes insulin resistance and hepatic fat accumulation [103].

Adipokines are directly transported to the liver via the portal vein, where they play a significant role in influencing liver diseases [104]. Different adipokines show different effects in the steatotic condition. For example, adiponectin prevents inflammatory responses in NAFLD. The dysregulation of adipokines may participate in the pathogenesis of NAFLD in poultry (Table 3) [105,106].

Leptin excreted by adipocytes, a major signaling molecule earliest to be discovered correlated with development of fatty liver condition in poultry, plays a role in regulating energy balance and body weight [107]. In poultry, elevated leptin levels may imply a higher risk for NAFLD, as fat accumulation in the liver may be due to the fact that the signals from the hormone leptin are not received anymore (leptin resistance) [108].

**Table 3 ijms-26-08460-t003:** Effect of different adipokines involved in NAFLD.

Adipokine	Effect	Reference
Leptin	In poultry, elevated leptin levels are associated with leptin resistance, a state in which hepatic and peripheral tissues fail to respond to leptin signaling. This impairment disrupts lipid homeostasis by reducing fatty acid oxidation while sustaining lipogenesis. Consequently, excess lipid deposition occurs in the liver, predisposing birds to NAFLD progression.	[107]
Adiponectin	Adiponectin significantly inhibits adipocyte development in chickens by downregulating key adipogenic transcription factors such as C/EBPα and FAS, while simultaneously upregulating lipolytic genes like ATGL and its receptor AdipoR1. This suppression of adipogenesis is further mediated through activation of the p38 MAPK/ATF-2 and TOR/p70S6K signaling pathways, collectively leading to reduced lipid accumulation and fat deposition in chickens’ hepatic tissues.	[109]
Visfatin	Visfatin protein expression is primarily localized around the central vein in hepatic lobules exhibiting mild steatosis. Elevated levels of Visfatin in serum and liver tissue appear early in the disease progression, suggesting its potential involvement in the pathogenesis of NAFLD.	[110]
Resistin	Resistin alters mitochondrial morphology, reduces mitochondrial content, and promotes lipid accumulation under high-fat diet conditions.	[111]
Chemerin	In chickens, chemerin is implicated in the control of lipid metabolism, exhibiting a negative correlation between its plasma levels and fattening, as well as hepatic expression. It has a modulatory role in hepatic lipid accumulation and an impact on the onset of fatty liver disease.	[105]
Tumor Necrosis Factor-α (TNF α)	TNF-α induces insulin resistance by disrupting post-receptor insulin signaling pathways, originates from adipose tissues, and acts as important cytokine in pro-inflammatory pathways.	[112]
Interleukin-6 (IL-6)	Elevated IL-6 levels activate hepatic inflammatory signaling pathways, disrupt lipid metabolism, and promote hepatic steatosis, thereby facilitating the NAFLD.	[112,113]

### 4.7. Role of Environmental Stress in NAFLD

Environmental stress significantly influences the onset and progression of NAFLD in poultry populations. The key environmental stressors often seen in poultry production come from heat stress, overcrowding, rearing system, and lighting.

Heat stress: Elevated ambient temperatures induce heat stress, leading to oxidative stress and metabolic disturbances in poultry [114]. Heat stress disrupts lipid metabolism, resulting in increased hepatic fat deposition [24]. Implementing effective cooling systems and ventilation is crucial in mitigating heat-induced NAFLD.

Overcrowding: High stocking densities cause stress, hormonal imbalances, and reduced physical activity in poultry [115]. These factors contribute to metabolic disruptions and increased risk of NAFLD [116]. Ensuring adequate space and environmental enrichment can alleviate stress and reduce disease incidence [117].

Rearing system: Cage production techniques remain the leading method for rearing laying hens in various countries around the world [118]. Many European countries had already banned cage systems but still in many Southern and Eastern European countries, over 60% of hens are kept in cages [119]. However, keeping birds in cages is considered against animal welfare, as it inhibits the natural behavior of hen. Research conducted in Queensland indicated that FLHS accounted for almost 40% of mortalities in commercial caged laying hens, with larger number of birds exhibiting an increased risk. Although overall mortality rates were comparable among cage, barn, and free-range systems, FLHS accounted for 74% of fatalities in caged birds, underscoring the impact of housing conditions and restricted mobility on hepatic health [15].

Lighting: The liver exhibits robust circadian regulation of its metabolic functions [120]. Prolonged or irregular light interferes with the expression of hepatic clock genes, including BMAL1, CLOCK, PER, and CRY, which play a role in the regulation of lipid metabolism. According to a study by Chen et al., blue LED light exposure in poultry has been associated with circadian rhythm disruption, leading to hepatic lipid accumulation and features of NAFLD. This effect is linked to reduced Kupffer cell activity and elevated inflammatory responses in the liver [121]. Disrupted circadian rhythms compromise lipid homeostasis, elevating the likelihood of hepatic fat formation [122,123]. Poultry exposed to prolonged lighting typically exhibit increased nocturnal eating, resulting in excessive caloric diet intake. Overconsumption, particularly during protracted photoperiods, may enhance lipogenesis, facilitating hepatic fat accumulation [124].

### 4.8. Influence of Production Phases on NAFLD

As laying hens age, hepatic function may decline due to an increased incidence of hepatic steatosis and fatty liver disease, which markedly impairs reproductive performance and reduces egg production [125]. The risk of NAFLD varies across different production phases.

Laying cycle (till 52 weeks): During this phase, hens experience heightened metabolic demands for egg production, leading to increased hepatic lipid synthesis. Without balanced nutrition and optimal environmental conditions, the risk of hepatic fat accumulation and NAFLD escalates [125].

Late-laying period: As the egg production period prolongs more than 52 weeks, the egg production capacity decreases due to decline in certain follicle stimulating hormones and number of primary follicle production [126]. However, residual hepatic lipid accumulation from the peak laying period can persist, maintaining the risk of NAFLD [127]. Careful management during this transition is essential to prevent disease progression.

### 4.9. Role of Gut Microbiota in NAFLD Development and Prevention

The gut microbiota plays an essential role in host metabolism, immune modulation, and liver function, making it a key factor in the pathophysiology of NAFLD. The poultry gastrointestinal tract hosts a diverse microbial ecosystem that influences nutrient absorption, bile acid metabolism, and systemic inflammation, all of which are closely linked to liver health.

Multiple studies on animal and human models generated evidence that NAFLD is associated with gut microbiota dysbiosis [128]. Dysbiosis or an imbalance in gut microbiota has been associated with increased intestinal permeability, endotoxemia, and lipid dysregulation, thereby contributing to the onset and progression of NAFLD [129]. Gut microbiota makeup ranges from basic steatosis to NASH, cirrhosis, and fibrosis. Consequently, gut microbiota may serve as indicators for the development and intensity of NAFLD [130,131]. Recent study showed for the first time that duodenal *Lactobacillus* emerged as the most important microbial factor compared to the microbiomes of other intestinal segments, potentially alleviating steatosis by boosting folate production and supporting the methionine cycle [60].

Mechanism of development of NAFLD in poultry is summarized in Figure 1.

## 5. Disease Diagnosis and Emerging Strategies to Prevent NAFLD in Poultry

NAFLD in poultry may exhibit a range of symptoms, which can be broadly categorized into production-related signs, behavioral alterations, and gross morphological changes [11]. The severity of these symptoms often correlates with the intensity of hepatic dysfunction. In poultry, NAFLD is characterized by excessive feed intake, abnormal adiposity leading to increased body weight, and a decline in reproductive performance, notably reduced egg production [12].

Key measurable biomarkers for NAFLD/FLHS in laying hens include elevated serum liver enzymes ALT and AST, which are consistently higher in affected birds and recommended for flock-level diagnosis, along with increases in serum triglycerides and total cholesterol that track risk and severity [132,133]. Endocrine status is also informative: circulating estradiol is elevated in hens with FLHS and experimentally precipitates steatosis/hemorrhage, supporting its use as a risk indicator [134,135]. Finally, dysregulation of the yolk-precursor pathway, vitellogenin, and VLDLy/apoVLDL-II, which are estrogen-induced hepatic products, reflects heightened yolk-lipid export from the liver and is perturbed in FLHS [136,137].

In case of severe disease, birds brought to necropsy commonly show extensive hemorrhaging and the presence of large blood clots within the abdominal cavity, which may lead to hepatic rupture [138]. Current diagnostic procedures for NAFLD in live birds include signs of decreased physical activity and increased feed intake. The emerging omics technologies, which include transcriptomics, proteomics, and metabolomics, expedite our knowledge acquisition about metabolic diseases, particularly NAFLD [139]. Most studies currently emphasize the role of lipid metabolism genes including peroxisome proliferator-activated receptors (PPARs) and liver X receptors (LXRs) within the progression of NAFLD in the liver [140]. The study of gene expression patterns in NAFLD-affected poultry may facilitate the finding of specific indicators for early diagnosis of NAFLD.

NAFLD is commonly observed in layer and breeder birds raised in commercial farming systems. Early diagnosis is imperative for successful intervention and management. Emerging studies are investigating the use of probiotics, prebiotics, and symbiotic to restore gut microbial balance and prevent hepatic steatosis. Prebiotic usage may lower lipids inside the hepatocellular membrane [141]. Manipulating gut microbiota composition through dietary and microbial interventions represents a promising avenue for prevention and control of NAFLD in poultry.

Omics-based methods enable complete transformations in the management to prevent NAFLD in poultry, since they reveal mechanisms of disease development while identifying treatment targets for NAFLD.

## 6. Knowledge Gaps and Future Directions in NAFLD Research in Poultry

NAFLD significantly affects poultry health, particularly among high-yield laying hens, but despite advances in understanding its pathophysiology, diagnosis, and management, critical gaps in knowledge remain. These gaps hinder the development of effective treatment and prevention strategies, which in turn limits the ability to achieve comprehensive economic control of NAFLD in commercial poultry. One of the primary limitations is the unclear understanding of the genetic and epigenetic factors that contribute to the disease’s susceptibility in poultry. While some progress has been made, this knowledge gap prevents the identification of at-risk populations and the development of targeted breeding programs to enhance resistance to NAFLD.

Additionally, most research is conducted in controlled environments, with a lack of large-scale, real-world field studies that could better reflect the complexities of commercial poultry systems. This limitation hampers the translation of laboratory findings into practical solutions for the industry. Research conducted in farm environments can reveal the role of environmental stress such as heat stress, overcrowding, and poor ventilation in the development of NAFLD. Laboratory studies often underestimate the impact of these stress factors, which are prevalent in commercial poultry production. By observing poultry in natural settings, researchers can identify the key risk factors contributing to NAFLD and assess the most effective strategies for preventing and managing the disease under realistic conditions.

There is a shortage of long-term data on the outcomes of interventions for NAFLD in poultry, with most studies focusing on short-term effects. Without extensive long-term research, it remains challenging to assess the sustainability and economic viability of these interventions. Long-term follow-up studies that monitor poultry flocks throughout their lifespan are essential to understanding the progression of NAFLD. Such studies would allow researchers to track how the disease develops over time in response to dietary and genetic changes, as well as the impact of environmental stress more precisely. Additionally, it would provide valuable data on whether specific interventions, such as dietary supplements or genetic selection programs, can reduce the prevalence and severity of NAFLD over extended periods. To generate robust findings, these studies should involve large, diverse poultry populations to accurately assess the effects of various interventions on disease progression and production outcomes.

To address these knowledge gaps, future research should focus on identifying genetic and epigenetic markers associated with NAFLD susceptibility, conducting large-scale field studies to evaluate management practices and interventions, and gathering long-term data on the effectiveness of treatments. By prioritizing these areas, researchers can develop more targeted and economically feasible solutions for controlling NAFLD in commercial poultry.

## 7. Conclusions

Researchers now study NAFLD in poultry because it threatens both their well-being and production performance while causing financial challenges to the industry. New research about gut bacteria and genomics assists our understanding of liver disease by providing better detection options. Our research still needs more information about how genetics and epigenetics affect liver disease responses to interventions. Current knowledge about NAFLD needs testing by field experiments to study these poultry systems more accurately.

Future research will succeed only through combined methods that study poultry genetics alongside diet patterns and environmental administration. These methods will help develop better long-term solutions to treat and stop NAFLD in poultry. Different scientific areas must work together to find new solutions for healthy poultry farming on an international scale.

## Figures and Tables

**Figure 1 ijms-26-08460-f001:**
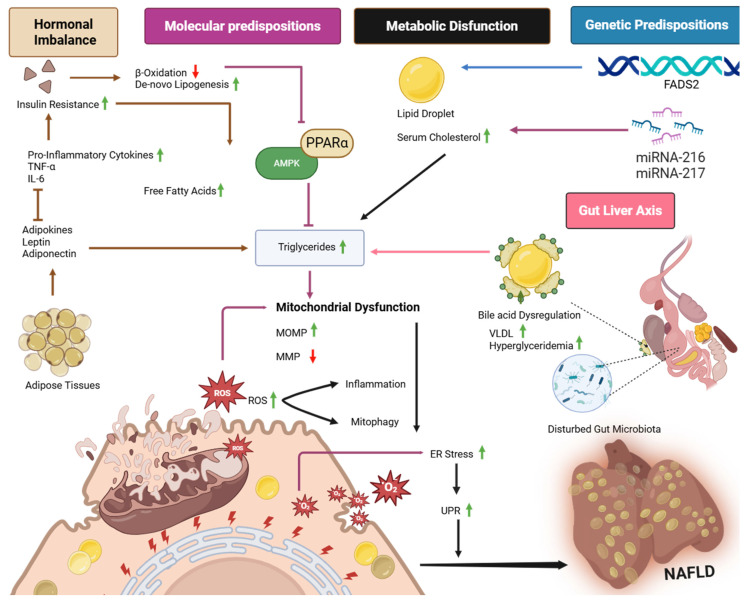
**Mechanism of development**. Insulin resistance upregulates De novo lipogenesis and downregulates β-oxidation, which leads to increase in free fatty acids and lipid droplet buildup in the liver. With the upregulated lipogenesis and decreased β-oxidation, the activation of AMPK and PPARα fails, which impairs lipid clearance and causes increase in triglyceride. Pro-inflammatory cytokines such as TNF-α and IL-6 exacerbate insulin resistance and lipid abnormalities. Disruptions in adipokines, including leptin and adiponectin, additionally facilitate a reduction in mitochondrial outer membrane permeability. Metabolic disturbances, such as increased blood cholesterol, VLDL, and hypertriglyceridemia, are exacerbated by bile acid imbalance. The gut–liver axis, characterized by a modified microbiota and elevated levels in the intestine, contributes to heightened triglyceride levels, resulting in systemic inflammation. Genetic factors, namely polymorphisms in the FADS2 gene, contribute to elevated blood cholesterol, hence disrupting bile acid homeostasis and causing mitochondrial dysfunction. MicroRNAs, including miRNA-216 and miRNA-217, are also involved in metabolic changes causing fibrosis of liver. Mitochondrial dysfunction, characterized by increased mitochondrial outer membrane permeability (MOMP) and disrupted membrane potential (MMP), is a central factor that results in impaired energy metabolism, elevated ROS production, and inflammation, while endoplasmic reticulum stress triggers UPR (unfolded protein response), exacerbates lipid accumulation, and hepatic inflammation.

**Table 2 ijms-26-08460-t002:** Unique aspects of avian lipid metabolism compared to mammals.

Feature	Poultry	Mammals
Lipoprotein transport	Portomicrons facilitate the direct transfer of lipids from the intestines to the liver through the portal vein, circumventing the lymphatic system [47].	Chylomicrons transport undigested fat- and fat-soluble vitamins via the lymphatic system to the blood stream [48].
Fatty acid composition	Polyunsaturated fatty acids (PUFA) can be synthesized by poultry birds, a characteristic unique to avian species [49].	Mammals are unable to synthesize PUFAs, and it is induced via diet [50].
Site of lipogenesis and storage	Liver is the primary site of lipogenesis and adipose tissues are the sites of storage in poultry [51].	De novo lipogenesis occurs in adipose tissues, and fat is stored in the liver [51,52]

## Data Availability

Data sharing is not applicable to this article as no datasets were generated or analyzed in the current study.

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
