# Peer review of "Non-Alcoholic Fatty Liver Disease in Poultry: Risk Factors, Mechanism of Development, and Emerging Strategies"

_ijms, 2025, doi:10.3390/ijms26178460_

Round 1

Reviewer 1 Report

Comments and Suggestions for Authors

The article “Non-Alcoholic fatty Liver Disease in Poultry: Risk Factors, Mechanism of Development, and Emerging Strategies” provides an excellent synthesis of the poultry NAFLD mechanism, covering genetics, metabolism, and environment, and addresses key potentially transformative issues in poultry production. However, there are grammatical errors, redundant and confusing words, and need to differentiate the NAFLD between poultry and mammalian. This manuscript is interesting and can be accept after further revision. Details as follows:

Abstract

“Compromises not only bird welfare but also productivity” Rephrase for parallel structure: “compromises bird welfare, productivity, and economic sustainability.”

“An integrative framework is an advocated one”. Suggest: “An integrative framework is advocated, synergizing genetics, nutrition...”

“Hormonal imbalances” is recommended to change to “Hormonal dysregulation” or “Endocrine disturbances”, to accurately describe abnormal hormone levels. Keep consistent in this manuscript.

Graphical Abstract

How to understand the use of facilitation symbols (red arrows) and inhibition symbols (green). Need to give more explanation.

Introduction

Line 7: “world malnutritional problems” Replace with “global malnutrition problems.”

  1. Mechanisms of Fatty Liver Development in Poultry

Sections 4.1 and 4.6 both deal with disorders of bile acid metabolism, and it is recommended to integrate or clearly distinguish them.

4.2. Molecular Basis of NAFLD Development in Poultry

“miR216 and miR217 are the small noncoding RNA molecules” Replace with “miR-216 and miR-217 are small noncoding RNAs.”

4.6. Role of Environmental Stress in NAFLD

Part Lighting, unable to determine if light is associated with NAFLD, please provide references.

  1. Knowledge Gaps and Future Directions in NAFLD Research in Poultry

“Non-alcoholic fatty liver disease” uses abbreviations, please.

Table 1

“Poor Health Condition” may be “hepatic steatosis”? Maintain consistent word usage throughout the manuscript.

For part “Gut Microbiota”, please give concise descriptions to facilitate reading and avoid colloquial remarks.

Table 3

Corrected header format.

“Adiponectin significantly inhibits adipocyte development, hence reducing fat accumulation in chickens.” Statement is not clear enough.

Confuses the research on NAFLD in mammals (e.g., mice, humans) and poultry, we should focus on poultry as the title is related with poultry.

Figure 1

Add figure legend, please.

Provide poultry liver image instead of mammals.

The signaling pathways described in Section 4.2 do not appear to be shown.

Also, be careful with manuscript format specifications such as L83, L180, L232, L367.

Author Response

The article “Non-Alcoholic fatty Liver Disease in Poultry: Risk Factors, Mechanism of Development, and Emerging Strategies” provides an excellent synthesis of the poultry NAFLD mechanism, covering genetics, metabolism, and environment, and addresses key potentially transformative issues in poultry production. However, there are grammatical errors, redundant and confusing words, and need to differentiate the NAFLD between poultry and mammalian. This manuscript is interesting and can be accept after further revision. Details as follows:

Response: We sincerely thank the reviewer for their insightful comments and valuable suggestions. We have revised the manuscript accordingly and believe the changes have significantly improved the overall quality and clarity of the paper. Below, we provide a point-by-point response to each comment.

Abstract: “Compromises not only bird welfare but also productivity” Rephrase for parallel structure: “compromises bird welfare, productivity, and economic sustainability.”

Response: Thank you very much for the suggestion. We have rephrased the sentence as suggested. Please see L16-17.

“An integrative framework is an advocated one”. Suggest: “An integrative framework is advocated, synergizing genetics, nutrition...”

Response: We appreciate your suggestion and changes have been made accordingly. Please see L26-L27.

“Hormonal imbalances” is recommended to change to “Hormonal dysregulation” or “Endocrine disturbances”, to accurately describe abnormal hormone levels. Keep consistent in this manuscript.

Response: We appreciate this observation and changed Hormonal Imbalance to Hormonal Dysregulation. Please see L314.

Graphical Abstract

How to understand the use of facilitation symbols (red arrows) and inhibition symbols (green). Need to give more explanation.

Response: Thank you for highlighting this important point. We have now added a detailed explanation of the facilitation and inhibition symbols used in our figures. Specifically, red arrows indicate facilitation or activation of a pathway or process, whereas green blunt-ended lines indicate inhibition or suppression. We have provided more text in the graph to explain the symbols.

Introduction

Line 7: “world malnutritional problems” Replace with “global malnutrition problems.”

Response: Thank you for pointing out this alternative suggestion of phrase and we have changed it accordingly. Please see L74.

Mechanisms of Fatty Liver Development in Poultry

Sections 4.1 and 4.6 both deal with disorders of bile acid metabolism, and it is recommended to integrate or clearly distinguish them.

Response: Thank you for your thoughtful observation. We would like to clarify that Section 4.1 discusses the role of hepatic lipid accumulation and bile acid dysregulation in the pathogenesis of NAFLD, focusing on internal metabolic dysfunction. In current manuscript, Section 4.7 (previously 4.6) addresses the impact of environmental stress on NAFLD, which may secondarily affect bile acid metabolism through stress-induced physiological changes but no findings regarding this impact has been reported in this manuscript.

Additionally, we believe there may have been some confusion with Section 3, which outlines the normal physiological process of bile acid metabolism in poultry, serving as a background for understanding the disruptions discussed in current manuscript Section 4.2 (previously 4.1).

This is our point of view to clarify difference between headings. We look forward for your further response if your concerns are clarified or we can make more efforts to resolve it.

Molecular Basis of NAFLD Development in Poultry

“miR216 and miR217 are the small noncoding RNA molecules” Replace with “miR-216 and miR-217 are small noncoding RNAs.”

Response: Thank you for your keen observation. The writing format for small noncoding RNAs has been changed according to your suggestions. Please see L240.

4.6. Role of Environmental Stress in NAFLD

Part Lighting, unable to determine if light is associated with NAFLD, please provide references.

Response: Thank you for the insightful comment. As discussed in  current manuscript section 4.7 (previously section 4.6), irregular lighting conditions interfere with specific genes and dysregulate lipid metabolism. Moreover considering your comment, we have also added a more clear insight on this matter by adding another research reference conducted by Chen et al., 2023, which clarifies that LED light exposure is associated with disruption in circadian rhythms and ultimately a reason for onset of NAFLD in poultry and elevates the inflammatory response. Please see L379-382.

The access link for this particular study is also provided for a more in-depth study

Chen, Y.-T., Huang, P. Y., Chai, C.-Y., Yu, S., Hsieh, Y.-L., Chang, H.-C., Kuo, C.-W., Lee, Y. C., & Yu, H.-S. (2023). Early detection of the initial stages of LED light-triggered non-alcoholic fatty liver disease by wax physisorption kinetics-Fourier transform infrared imaging. Analyst, 148(3), 643–653.https://doi.org/10.1039/d2an01546c

Knowledge Gaps and Future Directions in NAFLD Research in Poultry

“Non-alcoholic fatty liver disease” uses abbreviations, please.

Response: We appreciate your suggestion. The word Non-alcoholic fatty liver disease has been replaced with the abbreviation NAFLD to maintain a constant flow throughout the whole manuscript.

Table 1

  1. “Poor Health Condition” may be “hepatic steatosis”? Maintain consistent word usage throughout the manuscript.

Response: Thank you for your keen observation. This risk factor aims to discuss about poultry birds already health compromised due to metabolic disorders,  are more prone to get NAFLD due to weakened immunity. This susceptibility could be due to alteration in diet formulation, nutritional cause or environmental factor. Moreover the phrase Poor Health Condition has been replaced with “Prolonged Metabolic Disease”  to avoid confusion of concept for readers.

  1. For part “Gut Microbiota”, please give concise descriptions to facilitate reading and avoid colloquial remarks.

Response: Thank you for the comment. We have rephrased and made necessary changes in this part to avoid vague remarks and facilitate the reader with a concise and comprehensive text.

Table 3

  1. Corrected header format.

Response: Thank you for your deep observation. We have corrected the header format by removing unnecessary prepositions and according to the header format of Table1 and Table2.

  1. “Adiponectin significantly inhibits adipocyte development, hence reducing fat accumulation in chickens.” Statement is not clear enough.

 Confuses the research on NAFLD in mammals (e.g., mice, humans) and poultry, we should focus on poultry as the title is related with poultry.

Response: We appreciate your insightful comment. The statement has been taken from a research conducted by J.Yan et al; 2014, specifically on chicken model and resulting inconclusion that adiponectin has a remarkable effect on impairment of adipocyte differentiation, which contributes to the negative regulation of fat deposition in chicken. Moreover we have make this statement more clear and elaborative to avoid conflict of statement with research in human and mice (mammals) NAFLD.

 For more detailed overview access links are also provided below

  1. Yan, J and Yang, H and Gan, L and Sun, C. Adiponectin-Impaired Adipocyte Differentiation Negatively Regulates Fat Deposition in Chicken. Anim. Physiol. Anim. Nutr. (Berl). 2014, 98, 530–537, doi:doi.org/10.1111/jpn.12107.
  2. Yan, J., Gan, L., Chen, D., & Sun, C. (2013). Adiponectin impairs chicken preadipocytes differentiation through p38 MAPK/ATF-2 and TOR/p70 S6 kinase pathways. PLoS One8(10), e77716. doi: 10.1371/journal.pone.0077716.

Figure 1

 Add figure legend, please. Provide poultry liver image instead of mammals. 

The signaling pathways described in Section 4.2 do not appear to be shown.

Response: We appreciate you deep consideration regarding information presented in figures. We have now added the molecular mechanism in picture depicting section 4.3 in current manuscript (previously section 4.2). however, to avoid any confusion the arrows carring out a whole mechanism are changed according to the colour of corresponding heading. The small green arrows pointing upwards shows increase while the small red arrows represent the decrease in expression. The abbrevations used in picture are also clearly adressed in captian of picture below. We appreciate you for highlighting specific image of liver which has been now replaced with poultry liver. 

Reviewer 2 Report

Comments and Suggestions for Authors

For a scientific review addressing the underlying mechanisms by which hepatic lipid accumulation, metabolic dysfunctions, hormonal imbalances, genetic susceptibilities, and environmental stress contribute to the development of NAFLD to have an impact and be publishable in high-quality scientific journals, it is essential to meet internationally recognized methodological standards.

These include clearly defining the objective and designation as a review, preferably as a systematic review, with an explicit and reproducible methodology that details systematic literature searches in major databases and pre-established inclusion/exclusion criteria.

Adopting the PRISMA guidelines ensures transparency, including a flow diagram documenting study selection and rigorous reporting standards. Evaluating study quality using tools like GRADE adds validity and robustness to findings.

Results should be presented systematically to highlight pathophysiological patterns, clinical relationships, and knowledge gaps. The discussion must critically address limitations and heterogeneity and propose directions for future research or clinical applications.

Following these guidelines will transform the review into a reliable reference that informs clinicians and researchers, enhancing decision-making and fostering new studies on NAFLD.

Author Response

For a scientific review addressing the underlying mechanisms by which hepatic lipid accumulation, metabolic dysfunctions, hormonal imbalances, genetic susceptibilities, and environmental stress contribute to the development of NAFLD to have an impact and be publishable in high-quality scientific journals, it is essential to meet internationally recognized methodological standards.

These include clearly defining the objective and designation as a review, preferably as a systematic review, with an explicit and reproducible methodology that details systematic literature searches in major databases and pre-established inclusion/exclusion criteria.

Adopting the PRISMA guidelines ensures transparency, including a flow diagram documenting study selection and rigorous reporting standards. Evaluating study quality using tools like GRADE adds validity and robustness to findings.

Results should be presented systematically to highlight pathophysiological patterns, clinical relationships, and knowledge gaps. The discussion must critically address limitations and heterogeneity and propose directions for future research or clinical applications.

Following these guidelines will transform the review into a reliable reference that informs clinicians and researchers, enhancing decision-making and fostering new studies on NAFLD.

Response: We sincerely thank the reviewer for the thoughtful and detailed feedback. We fully agree that reviews should follow rigorous methodological standards, and we appreciate the emphasis on clarity, transparency, and reproducibility.

 However, we respectfully note that the current review is structured as a narrative synthesis, rather than a systematic review. This approach was chosen deliberately due to the limited and heterogeneous body of literature on NAFLD in poultry, which does not yet support a fully systematic approach (e.g., meta-analysis or quantitative grading via GRADE and PRISMA).

To address the reviewer's concern, we clearly define the review as a narrative review.

Emphasize that this review aims to summarize current knowledge, identify emerging pathophysiological patterns, and highlight key research gaps relevant to poultry models of NAFLD.  We believe this framework provides a valuable reference for future research and will support more structured evidence syntheses as the field evolves.

Moreover, we have also added an additional section of methods at the end of introduction part pointing the sources of literature finding, overview, data collection methodology and ensure complete transparency of valid data by two researchers independently to avoid reporting conflicted data.

We have also added a brief statement in the conclusion part regarding limitations of the current evidence base and the potential for more systematic reviews once more original research becomes available.

Additionally, a framework of key targets of this narrative reporting is also provided below to satisfy your concerns regarding hypothesis and aim of the current review:

AIM

Due to the limited number of studies specifically addressing NAFLD in poultry provide an integrative framework combining genetic, nutritional, and environmental approaches to enhance poultry health, welfare, and productivity through effective NAFLD management.

Methodology

Comprehensive search of the literature by using web pages like Google Scholar, PubMed, science websites etc and for specific statistical data official websites of FAO, USDA etc.

For findings in field of poultry science we use specific keywords like non-alcoholic fatty liver disease; lipid accumulation; nutritional intervention; genetic selection; environmental stress; laying hen.               

Framework of narrative review paper

ABSTRACT

INTRODUCTION

LIPID METABOLISM IN POULTRY

BILE ACID METABOLISM IN POULTRY

MECHANISM OF FATTY LIVER DEVELOPMENT IN POULTRY

  1. Role of nutritional Imbalance in Development of NAFLD
  2. Role of Hepatic Lipid Accumulation and Bile Acid Dysregulation in NAFLD
  3. Molecular Basis of NAFLD Development in Poultry
  4. Molecular Markers in NAFLD in Poultry
  5. Signaling pathways
  6. Signaling pathways

  1. The Role of ER-Mitochondrial Interplay in NAFLD
  2. Genetic Predispositions of NAFLD in Poultry
  3. Role of Hormonal Dysregulation in Progression of NAFLD
  4. Role of Environmental Stress in NAFLD
  5. Influence of Production Phases on NAFLD
  6. Role of Gut Microbiota in NAFLD Development and Prevention

DISEASE DIAGNOSIS and EMERGING STRATEGIES to PREVENT NAFLD in POULTRY

KNOWLEDGE GAPS and FUTURE DIRECTIONS in NAFLD RESEARCH in POULTRY

CONCLUSIONS

We sincerely appreciate the reviewer’s insightful comments, which have greatly improved the quality and clarity of our manuscript. We hope that we are able to satisfy your concerns, and We are looking forward to hearing from you at your convenience.

Reviewer 3 Report

Comments and Suggestions for Authors

Brief Summary

This paper reviews the impact of NAFLD in poultry, particularly in laying hens. This is a very interesting topic that brings a new perspective to poultry production.

The study is generally well written in a correct English. The presentation of tables and figures deserves to be reviewed.

General Concept

The paper is rather well structured but would sometimes deserve more precise recommendations.

My main criticism concerns the nutritional assessment, particularly the biomarkers of interest, as well as the absence of nutritional recommendations to be made at the moment.

For example,fhe impact of choline and methionine are only briefly mentioned (line 150). However, alterations in the metabolism of methionine and S-adenosylmethionine (SAMe) are very important in the establishment of NAFLD, at least in mammals. Can you expand on this point in poultry?

I think that your excellent review is missing also a point on the strategies implemented in farms to combat NAFLD, particularly on a nutritional level, but not only.

Specific comments referring to line numbers,

The formatting of the tables generally makes it difficult to read the paper. Can you improve this point?

Line 33 : we don't really see what this graph is doing here, I would suggest to put it at the end of the intro

Line 91 : Can we have a real idea of the economic impact of the NAFLD in poultry?

Line 234 : The impact of genetic lineage on NAFLD is really very intersting. Can you expand on this part?

Line 355 : This part seems a little too vague to me. Can we have a more precise idea of the biomarkers to target?

Author Response

This paper reviews the impact of NAFLD in poultry, particularly in laying hens. This is a very interesting topic that brings a new perspective to poultry production.

The study is generally well written in a correct English. The presentation of tables and figures deserves to be reviewed.

Response: We would like to express our sincere gratitude to the reviewer for their thoughtful and constructive feedback. We have carefully addressed each comment and made the necessary revisions, which we believe have enhanced the scientific rigor and clarity of the manuscript. Our detailed responses to each point are provided below.

General Concept

The paper is rather well structured but would sometimes deserve more precise recommendations.

My main criticism concerns the nutritional assessment, particularly the biomarkers of interest, as well as the absence of nutritional recommendations to be made at the moment.

For example, the impact of choline and methionine are only briefly mentioned (line 150). However, alterations in the metabolism of methionine and S-adenosylmethionine (SAMe) are very important in the establishment of NAFLD, at least in mammals. Can you expand on this point in poultry?

Response: Thank you for highlighting an important point related to nutrition that was missing in our manuscript. Considering the importance of dietary factors we have comprehensively added a new heading (section 4.1). We thoroughly studied the already published work on Amino acids specially role of SAMe in development of NAFLD in chicken models and found that Changes in methionine/SAMe metabolism, whether due to inadequate methionine intake, poor conversion to SAMe, or a lack of choline/methyl donors, are crucial in poultry non-alcoholic fatty liver disease (NAFLD) because they impair the liver's capacity to store and use fat.

Specific comments referring to line numbers

The formatting of the tables generally makes it difficult to read the paper. Can you improve this point?

Response: We appreciate your feedback on the table readability. The current tables follow the IJMS/MDPI formatting standards; however, we recognize that the sentence wrapping and cells pacing could affect readability. We have revised the tables to improve clarity by reducing excessive spacing, avoiding mid-word breaks, and ensuring concise phrasing within each cell. These adjustments maintain journal style while making the tables more readable.

Line 33: we don't really see what this graph is doing here, I would suggest to put it at the end of the intro.

Response: We appreciate your deep observation regarding the figure placement. The graph you referenced in line 33 is the graphical abstract, which IJMS formatting guidelines require to appear immediately after the main abstract, before the introduction. We have clarified in our revision that this is the graphical abstract and ensured it is correctly labeled and placed according to journal requirements to avoid confusion.

Line 91: Can we have a real idea of the economic impact of the NAFLD in poultry?

Response: Thank you for highlighting this point and we appreciate your comment on real world economic analysis impacted due to NAFLD in poultry sector. Since to-date there is no sufficient data available on economic statistics on NAFLD specifically in poultry. Based on already published literature on this topic we have added a brief overview (L58-70) highlighting prevalence, mortality and loss in productivity due to reported outbreaks.

Line 234: The impact of genetic lineage on NAFLD is really very interesting. Can you expand on this part?

Response: Thank you for your deep interest in the genetic factors influencing NAFLD. We have expanded the corresponding section 4.5 (L297-308) to include evidence demonstrating breed-specific predisposition to hepatic steatosis in poultry, including paternal transmission of fatty liver traits, heritability of steatosis traits, and key genes implicated across breeds (e.g., ACACA, FABP1, APOA4, and ARAP2). These additions, supported by transcriptome and genomic analyses in chickens, underscore the critical role of genetic lineage in NAFLD susceptibility. Citations have also been added accordingly.

Line 355: This part seems a little too vague to me. Can we have a more precise idea of the biomarkers to target?

Response: We appreciate the reviewer’s suggestion regarding a clear explanation of biomarkers to target while diagnosing NAFLD in poultry. In the revised manuscript, we clearly outline that these include: (1) elevated serum alanine aminotransferase (ALT) and aspartate aminotransferase (AST) levels, which are indicative of hepatocellular damage; (2) increased serum triglycerides and total cholesterol, reflecting impaired lipid metabolism; (3) an elevated liver-to-body-weight ratio, signifying hepatomegaly; (4) higher circulating estrogen concentrations, which have been linked to enhanced hepatic lipid deposition; and (5) abnormal yolk precursor profiles, particularly altered vitellogenin and very low-density lipoprotein (VLDL) levels, indicating disrupted hepatic lipid synthesis and export. These biomarkers provide a more precise framework for both clinical assessment and experimental evaluation of NAFLD in poultry. Please see (L447-455).

Round 2

Reviewer 2 Report

Comments and Suggestions for Authors

The authors have implemented modifications to the manuscript; however, it still lacks the necessary methodological and formal structure to be considered a fully sound scientific article. While the efforts toward improvement are acknowledged, shortcomings remain that limit its academic consistency.

Therefore, the final decision regarding the suitability of its publication lies with the editor, who should assess whether, in its current form, this manuscript meets the minimum scientific quality standards required for acceptance as a review.

Academic Editors' Response

Thank you very much for your thoughtful and detailed review of the manuscript. Your comments regarding the limitations in originality were carefully considered during the editorial decision process. While we fully respect your evaluation, the editorial decision ultimately took into account the thorough revisions submitted by the authors and the positive assessments from the other reviewers. We felt the manuscript, in its revised form, offers sufficient scientific merit and utility to the IJMS readership, particularly through its species-specific synthesis and identification of research gaps. We greatly appreciate your contribution to the peer review process, and thank you again for your time and expertise.